# Lessons from 20 years of medical cannabis use in Canada

**Minsup Shim[1], Hai Nguyen[2], Paul Grootendorst**[1,3] *

**1** Leslie Dan Faculty of Pharmacy, University of Toronto, Toronto, Ontario, Canada, **2** Faculty of Pharmacy, Memorial University of Newfoundland, St John's, Newfoundland and Labrador, Canada, **3** Department of Economics, McMaster University, Hamilton, Ontario, Canada

* paul.grootendorst@gmail.com

**Data Availability Statement:** The data are available at: Grootendorst, Paul, 2023, "Replication Data for: Canada medical cannabis access regime administrative data and licensed producer catalog data", https://doi.org/10.5683/SP3/SXSYPD, Borealis

## Abstract

### Background

Canada was one of the first countries to regulate the medical use of cannabis. However, literature on Canada's medical cannabis program is limited.

### Methods

We use administrative data from the medical cannabis program, and licensed cannabis vendor catalog data to describe a) the participation of patients, physicians, and cannabis vendors in the program from its inception in 1999 to 2021, and b) trends in medical cannabis consumption, prices and potency. We also use national surveys conducted over the last several decades to estimate trends in regular cannabis use (medical or otherwise) and how it changed during the medical cannabis access regimes.

### Results

In 2001, the Canadian government granted access to those with physician-documented evidence of a severe health problem that could not be managed using conventional therapies. Most patients accessed cannabis grown under a personal production license. By 2013, authorized daily cannabis dosages were very high. In 2014, the government, concerned over illegal diversion, required that cannabis be purchased from a licensed commercial grower; personal production was banned. Physicians were given responsibility for authorizing patient access. To fill the regulatory void, the physician regulatory bodies in Canada imposed their own prescribing restrictions. After these changes, the number of physicians who were willing to support patient cannabis use markedly decline but the number of patients participating in the program sharply increased. Medical cannabis use varied by province–rates were generally lower in provinces with stricter regulations on physician cannabis prescribing. Most varieties of cannabis oil available for sale are now high in CBD and low in THC. Dry cannabis varieties, conversely, tend to be high in THC and low in CBD. Inflation adjusted prices of most varieties of medical cannabis have declined over time. We find that rates of daily cannabis use (medical or otherwise) increased markedly after the 2014 policy regime. The fraction of Canadians using cannabis daily increased again after the

**Funding:** The author(s) received no specific funding for this work.

**Competing interests:** The authors have declared that no competing interests exist.

2018 legalization of recreational cannabis; at the same time, participation in the medical access program declined.

## Conclusion

The implications for patient health outcomes of changes in the medical cannabis program and legalization of recreational use remains an important area for future research.

## Introduction

Canada introduced a medical cannabis access program in 1999, making it one of the first countries to do so. The regulations that govern patient eligibility and the legal sources of supply, however, have evolved over time. These changes have broadened patient eligibility criteria, made it easier for patients to obtain authorization, and created a regulated domestic cannabis industry which offers a large range of products that vary in both potency and dosage forms. In this paper, we provide descriptive evidence on the participation of physicians, patients and cannabis vendors in the Canadian medical cannabis access regime over the last two decades. As far as we are aware, we are the first to do so.

We begin by outlining the history of Canada's medical cannabis access regime. Next, we present empirical evidence on national trends in patient and physician participation in the medical cannabis regime. We also examine trends in the prices and varieties of medical cannabis for sale by authorized vendors, known as Licensed Producers. We then examine provincial trends in the volume of medical cannabis sold by Licensed Producers and explore possible reasons for the interprovincial variation. One such reason is the degree of restrictiveness of province-specific regulations on physician cannabis prescribing. Next, we use repeated cross sectional national level surveys to examine the possible effects of the liberalization of the medical cannabis access system on overall rates of regular cannabis use (medical or otherwise) in the community.

The plan of the paper is as follows. In the following section we briefly describe Canada's cannabis access policy over the last two decades. Next, we present the data used to describe various aspects of the program. We then present results and finally discuss results.

## Canada's Medical Cannabis Regulatory Policy, 1999–2021

Prior to 2001, severely ill patients could petition the federal Minister of Health for an exemption from the Controlled Drugs and Substances Act (CDSA); the CDSA prohibits or regulates the use of cannabis, opiates, and other drugs [1]. This ministerial exemption, codified in Section 56 of the CDSA, was unused until the Ontario Superior Court instructed Health Canada (the federal Ministry of Health) to liberalize medical cannabis access [2]. In response, the Minister granted authorization, on compassionate grounds, to two individuals who were living with HIV/AIDS [2]; these individuals were allowed to cultivate their own cannabis [3].

These regulations were overturned by the courts in 2000. A man who had experienced severe epilepsy since childhood, Terrance Parker, found that smoking cannabis was more effective than conventional medicines in reducing the frequency of debilitating seizures. He cultivated and used cannabis without ministerial approval and, in 1996, was charged under the CDSA. Parker prevailed at trial; the crown appealed and the court again ruled in favor of Parker. Moreover, the appeals court found that the arbitrariness of the Section 56 exemption breached patients' constitutional right to life, liberty, and security [4].

Health Canada duly promulgated the Marihuana Medical Access Regulations (MMAR) in 2001. Patients were required to apply to Health Canada for authorization to possess cannabis; the 29 page application required *inter alia* written documentation from the patient's primary physician (and in some cases a physician specialist as well) that the patient was either receiving palliative care or was experiencing debilitating symptoms related to specific medical conditions or treatments, such as severe nausea from chemotherapy, that could not be treated using conventional pharmaceuticals [2]. The first patients were enrolled in the program in early 2003.

Under the MMAR, authorized patients or their designates could obtain a license to grow cannabis plants; patients could also purchase dried cannabis from Health Canada at $5 per gram. These production licenses stipulated the number of cannabis plants allowed; this was determined formulaically from the authorized daily consumption and an estimated average yield per cannabis plant [5].

By December 2013, the average authorized daily consumption under the MMAR was about 18 grams. A person authorized to consume 18 grams daily was allowed to grow 88 cannabis plants [1]. A delegated producer could grow cannabis on behalf of 4 authorized users; thus a delegated production facility could contain several hundred plants [1]. Authorities grew concerned about both the safety of residential cannabis cultivation [6] and the diversion of cannabis to the illicit market [1]. These concerns were especially pronounced in the province of British Columbia. In that province, over 2 million cannabis plants were authorized under the MMAR.

In response to these concerns, Health Canada replaced the MMAR with a new set of regulations known as the Marihuana for Medical Purposes Regulations (MMPR). The MMPR, introduced in late 2013 and fully enacted on April 1, 2014, revoked the cannabis production licenses. Authorized patients could obtain cannabis (in dried form only) via mail order from a Licensed Producer, a commercial vendor approved and regulated by Health Canada.

The second goal of the MMPR was to remove Health Canada from the business of authorizing patient access to medical cannabis. Health Canada delegated this responsibility to physicians and nurse practitioners, who could authorize cannabis use through a one-page prescription-like form called a "medical document". Health Canada did not regulate the potency or daily consumption of cannabis, although it did set a possession limit of 150 grams. To fill the regulatory void [7–9], each provincial College of Physicians and Surgeons (hereafter, the "Colleges") imposed their own prescribing restrictions. (These Colleges, which are responsible for licensing, disciplining, and regulating physicians, are the provincial equivalents to state medical boards in the United States.) The cannabis prescribing rules varied by province and include *inter alia* requirements that the physician meet with the patient at least every 3 months, assess patient addiction risk, and in Quebec, enroll the patient into a research study on cannabis use [10]. None of the authorities that regulate nurse practitioners have allowed cannabis prescribing.

Patient groups, in turn, successfully petitioned the courts to allow the sale of cannabis oil and fresh cannabis leaves (in 2015) [11] and again allow patients (or their designates) to cultivate cannabis for personal use (in August 2016) albeit with more restrictions than under the MMAR. The revised set of rules are known as the Access to Cannabis for Medical Purposes Regulations (ACMPR) [12, 13].

More recently, in October 2018, Canada legalized the recreational use of cannabis for those aged 19 and older. Canada was only the second country, after Uruguay, to do so. As before, only Licensed Producers are eligible to produce cannabis commercially although individuals are allowed to grow a limited number of cannabis plants for personal use. Each province has authority over i) cannabis use, including age restrictions on users and locations that cannabis can be consumed; ii) retail sales formats; iii) excise taxes levied on cannabis sales; and finally, iv) the regulations governing cultivation for personal use. Recreational cannabis sales are

subject to both federal and provincial excise and sales taxes. Controversially, these taxes also apply to medical cannabis use. Fig 1 summarizes the timeline of medical and non-medical cannabis access regimes.

## Methods

We collected data that reflect physician and patient participation in the medical cannabis program, the prices and potency of medical cannabis sold by Licensed Producers, rates of medical cannabis use and the fraction of the population using cannabis regularly over time.

Information on the MMAR regime came from Health Canada administrative records and affidavits of a Health Canada employee [1, 14] in a Federal Court challenge over the constitutionality of the MMPR regime [15]. Health Canada provided data on the number of patients authorized to possess cannabis, by province and month, and the number of physicians supporting these cannabis possession applications nationally, by month. The affidavits contained data on the number of personal and delegated cannabis production licenses [14], and the number of cannabis plants authorized [1], by province, in December 2013. We expressed these data as rates per 100,000 population age 20+ using Statistics Canada provincial population estimates for July 2013 [16]. Some of the Health Canada administrative data we relied on are also available online [17, 18].

National data on the number of individuals authorized to possess cannabis, the number of production licenses, and permitted daily dosages, annually, from 2001–2014 were reported in an affidavit [14]. Daily dosages were reported in 18 irregularly spaced intervals, starting at 0 to 2.9 grams, then 3.0 to 4.9 grams, up to 250 to 299.9 grams. (Only 1 individual was authorized to use over 300 grams daily.) We used these data to estimate, by year, the average authorized daily dose; this is the weighted average of the interval midpoints, where the interval weights were the fraction of patients authorized to use the quantity of cannabis in the interval.

Health Canada also provided, for the MMPR/ACMPR regimes, data on the number of medical cannabis mail order shipments from Licensed Producers to authorized patients, by province and month. We expressed shipments as rates per 1,000 population, using Statistics Canada quarterly province-level population estimates [19].

Health Canada also provided data on the quantity of cannabis sold by Licensed Producers nationally, by presentation (dry vs oil) and month from May 2014 to September 2019. Dry cannabis includes "any part of a cannabis plant, except seeds, that has been through a drying process (dried flowers, pre-ground/milled (trim/shake), pre-rolls)" [20]. Oils are "made using extraction processing methods or by synthesizing phytocannabinoids and intended for inhalation or ingestion, including by absorption in the mouth or other routes of administration (e.g. vape pens, hash, tinctures, softgels, suppositories)" [20]. Dry cannabis is normally smoked and sometimes vaped, while oils are used primarily for vaping. Product sales were measured in kilograms until September 2018; afterwards cannabis oil was measured in litres. As a litre of oil weighs less than a kilogram, this measure change will cause unit oil sales to be modestly overstated.

Starting October 2019, Health Canada changed the way it reported Licensed Producer product sales. Sales volumes were measured in "packaged units" sold to patients, cannabis oil was relabelled as "cannabis extracts" and sales of cannabis "edibles" (which include cannabis-infused beverages, and foods such as gummies and chocolates) were reported. These data were obtained from the Government of Canada "Open Government" website [20].

We examined the impact on interprovincial rates of medical cannabis use of the number and type of cannabis prescribing restrictions imposed by each provincial College. To do so we surveyed these restrictions during April 2016 and assigned each restriction a score from 0.5 to

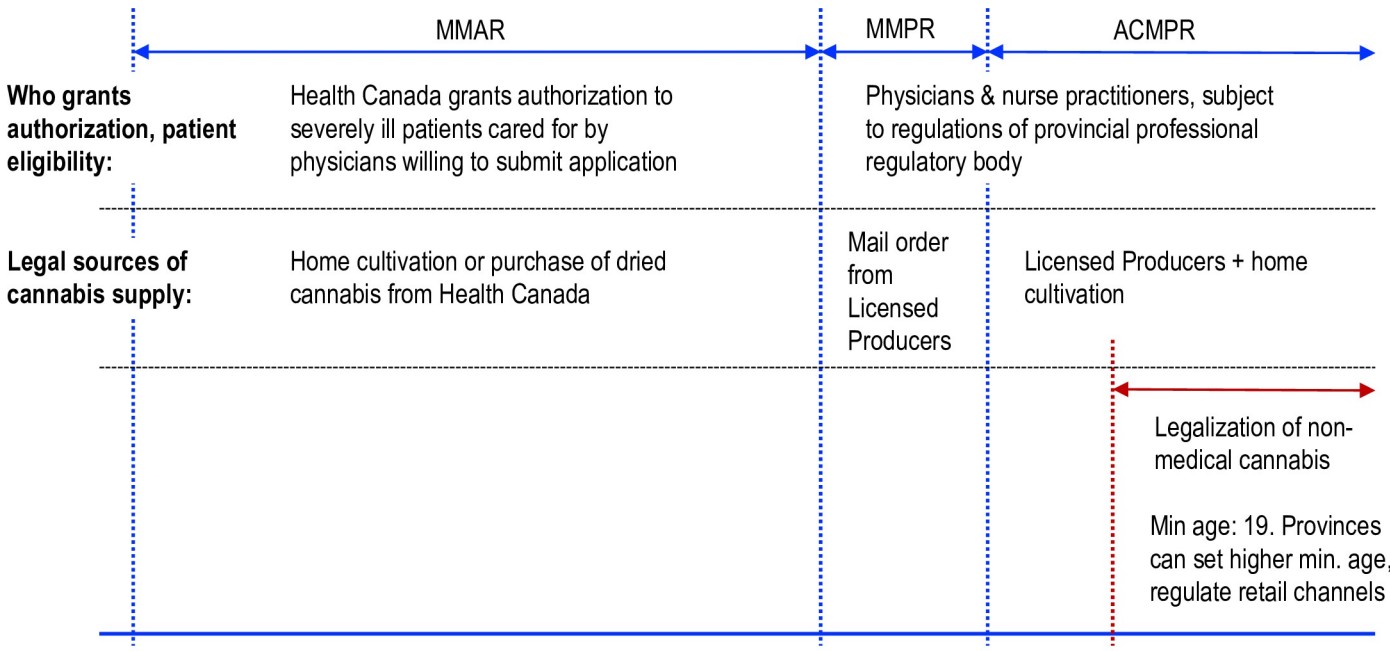

**Fig 1. Timeline of medical and recreational cannabis access regimes in Canada.** Note: MMAR: Marihuana Medical Access Regulations. MMPR: Marihuana for Medical Purposes Regulations. ACMPR: Access to Cannabis for Medical Purposes Regulations.

4, with larger values representing more onerous restrictions. For instance, the requirement that the prescriber meet with the patient in person to prescribe cannabis was assigned a score of 1. The requirement that the prescriber be the patient's primary physician was assigned a score of 3. These requirements were independently rated by two of the investigators and differences resolved via consensus. We then summed the scores to generate, for each province, a composite cannabis prescribing restriction score. These scores were plotted against province-level data on medical cannabis use. This was measured as the average number of cannabis shipments from Licensed Producers to registered clients over the three-month period March to May 2016. Rates were expressed as shipments per 100,000 population using the quarterly Statistics Canada population estimates [21].

We also investigated the association between the provincial restrictions on physician cannabis prescribing and the fraction of regular cannabis users (those who use cannabis weekly or daily) aged 20 and older who are enrolled in the medical cannabis program. This fraction was estimated as the ratio of the number of individuals enrolled in the medical access program in July 2015 to the number of regular users, estimated from the public use version of the 2015 Statistics Canada Canadian Tobacco, Alcohol and Drugs Survey. Presumably, the more stringent the regulations, the greater the fraction of regular users who obtain cannabis outside of the medical program.

To infer trends in the prices and potency of prescribed dry and oil-based cannabis under the MMPR/ACMPR regimes, we extracted data from the on-line product catalogs of Licensed Producers at the start of each month from June 2016 to December 2022 (except for December 2019). We excluded "micro-licence holders", Licensed Producers that are limited to a maximum of 200 square metres of growing surface. Despite comprising 38% of all federal licence holders by July 31, 2022, they accounted for less than 1% of LP sales [22].

We recorded the concentrations of the two primary active ingredients, tetrahydrocannabinol (THC) and cannabidiol (CBD) and the Canadian dollar price per gram (dried cannabis) or the price per ml (oil) for each product on offer. Products were assigned to one of four categories based on the strength of THC and CBD: THC≥10% and CBD≥10%; THC≥10% and CBD<10%; THC<10% and CBD≥10%; and THC<10% and CBD<10%. We calculated the average price of products, by strength category, month, and type of cannabis presentation (dry vs oil). We divided by the Statistics Canada all-item consumer price index (CPI) to adjust for inflation [23]. We did not collect pricing data for other types of cannabis products on offer, such as skin creams and other topicals or cannabis infused teas, chocolates and other types of edible cannabis.

These pricing data have some limitations. First, a referee notes that Licensed Producer reported concentrations of THC and CBD per ml of oil-based cannabis products can diverge from actual concentrations. This may introduce some error when estimating the price per unit of THC or CBD contained in the product. Second, to calculate average prices, ideally, we would weight the prices by the market share of the Licensed Producer selling the product, but Licensed Producer specific sales data are not publicly available. Third, some Licensed Producer pricing data was not available on-line; in some cases, on-line access was restricted to registered users.

To assess our price data, we compared our data with monthly Statistics Canada national "medicinal cannabis" CPI data from January 2019 to October 2022 [23]. These data are a weighted average of consumer price indices for both dry and oil forms [24]; each of the price indices are based on a sample of products [25]. Statistics Canada's publications do not disclose the weights in this weighted average or how cannabis product sampling was conducted.

We calculated the strength category specific share of all products on offer, by month, and type of cannabis presentation. We then graphed these data to illustrate trends in the potency of medical cannabis on offer. These graphs reflect trends in the potency of medical cannabis being used if the number of products in a strength category is proportional to the level of patient demand for all products in the category.

We next examined province-level trends in rates of regular cannabis use over the period 1989 to 2021 to assess possible impacts of the introduction of the medical cannabis access system in 2003 and its liberalization in 2013. To do so, we assembled the national health and drug use surveys conducted over this period and estimated the fraction of the residential population aged 20 and older that consumes cannabis daily or almost daily. These estimates were stratified by survey, year and province. Next, for each province, we fit a median spline curve with 9 bands to these year and survey-specific estimates to discern local trends. We use the public use versions of the 1989 National Alcohol and Drugs Survey; 1994 Canada's Alcohol and Other Drugs Survey; 2002 Canadian Community Health Survey; 2004 Canadian Addiction Survey (CAS); 2005–2013 Canadian Tobacco Use Monitoring Surveys; 2008–2012 Canadian Alcohol and Drug Use Monitoring Surveys (CADUMS); the 2013, 2015, 2017 Canadian Tobacco, Alcohol and Drugs Survey; the 2019–2021 Canadian Tobacco and Nicotine Survey; and the 2018, 2019 and 2020 National Cannabis Survey (NCS). All surveys were conducted by Statistics Canada except for the CAS, CADUMS and NCS; these were commissioned by Health Canada and conducted by private sector firms.

All these surveys used telephone interviews of provincial households with land-based telephones. Households were selected using random-digit dialing and questionnaires were administered to a randomly selected household member, age 15 or older, using Computer Assisted Telephone Interviewing. The NCS was the exception; households were selected using random-digit dialing of both land and cell phone numbers. Screening questions were asked over the phone and eligible individuals were invited to complete an on-line survey.

Households from the smaller provinces, and younger individuals were oversampled. The surveys include observation specific weights that adjust for household non-response and other factors that make population estimates consistent with known province-age-sex totals. We use these survey weights when estimating the proportion of the residential population aged 20 + that use cannabis regularly. The median number of observations per estimate was over 1,200; 75% of the estimates were obtained from samples exceeding 900 observations.

## Results

### Physician and patient participation in the medical access program

Panel (a) of Fig 2 plots the number of physicians across Canada who authorized medical cannabis use, by month, 2003–5 to 2014–3 (under the MMAR regime) and 2014–4 to 2021–12 (under the MMPR/ACMPR regime). Recall that under the MMAR, Health Canada adjudicated patient requests for medical cannabis. Under the MMPR and ACMPR regimes, Health Canada ceded prescribing authority to physicians. This policy change, and the attendant promulgation of provincial College regulations, evidently reduced the number of physicians participating in the medical access program. The policy change, however, no longer restricted medical cannabis to just severely ill patients; this may explain the increase in the number of patients registered in the program evident in panel (b) of Fig 2. At its peak, about 400,000 Canadians (just over 1% of the population) were registered. (An additional 40,000 individuals had licenses to cultivate cannabis under the ACMPR regime.) These two effects imply that the remaining participating physicians authorized cannabis use for a larger number of patients after the MMPR was introduced. Indeed, the number of patients per prescriber increased from about 5 under the MMAR regime to over 150 under the MMPR regime (panel c). It appears that this prescribing takes place mainly by physicians who work in "cannabis clinics" that specialize in cannabis authorizations, typically using online consultations [26]. In 2018, after the legalization of non-medical cannabis, the number of patients participating in the medical program declined.

### Cannabis dosage size, dosage form, and potency

The average daily dose of dry cannabis authorized by Health Canada under the MMAR increased gradually from 5.6 grams/day in 2001 to 8.6 grams/day in 2011. Mean dosages then increased sharply to 12.1 grams/day in 2012 and 20.6 grams/day in 2013 [14]. It is unlikely that MMAR program enrollees consumed all the authorized cannabis dosage amounts. This is unlikely given the evidence from law enforcement personnel that cannabis grown under the auspices of the MMAR was being diverted [1].

Most authorized patients elected to use cannabis grown under a personal or delegated production license. In 2013, only 19% of the 36,797 authorized users procured cannabis directly from Health Canada [14]. There are no data on the THC and CBD concentrations of cannabis authorized under the MMAR. While Health Canada regulated the number of cannabis plants individuals could grow, potency was unregulated.

Under the MMPR/ACMPR, physician-authorized daily doses declined over time, from 4 grams of dry cannabis in mid 2014 to 2 grams in late 2020. The average daily amount per Licensed Producer shipment was markedly lower. For instance, in the fourth quarter of 2016, the mean physician authorized daily dose was 2.5 grams but the average daily amount per Licensed Producer shipment was 0.79 grams [18]. This reduction in cannabis doses likely reflects the expansion of the medical access program to include patients who experienced lower levels of morbidity compared to those who were eligible under the MMAR.

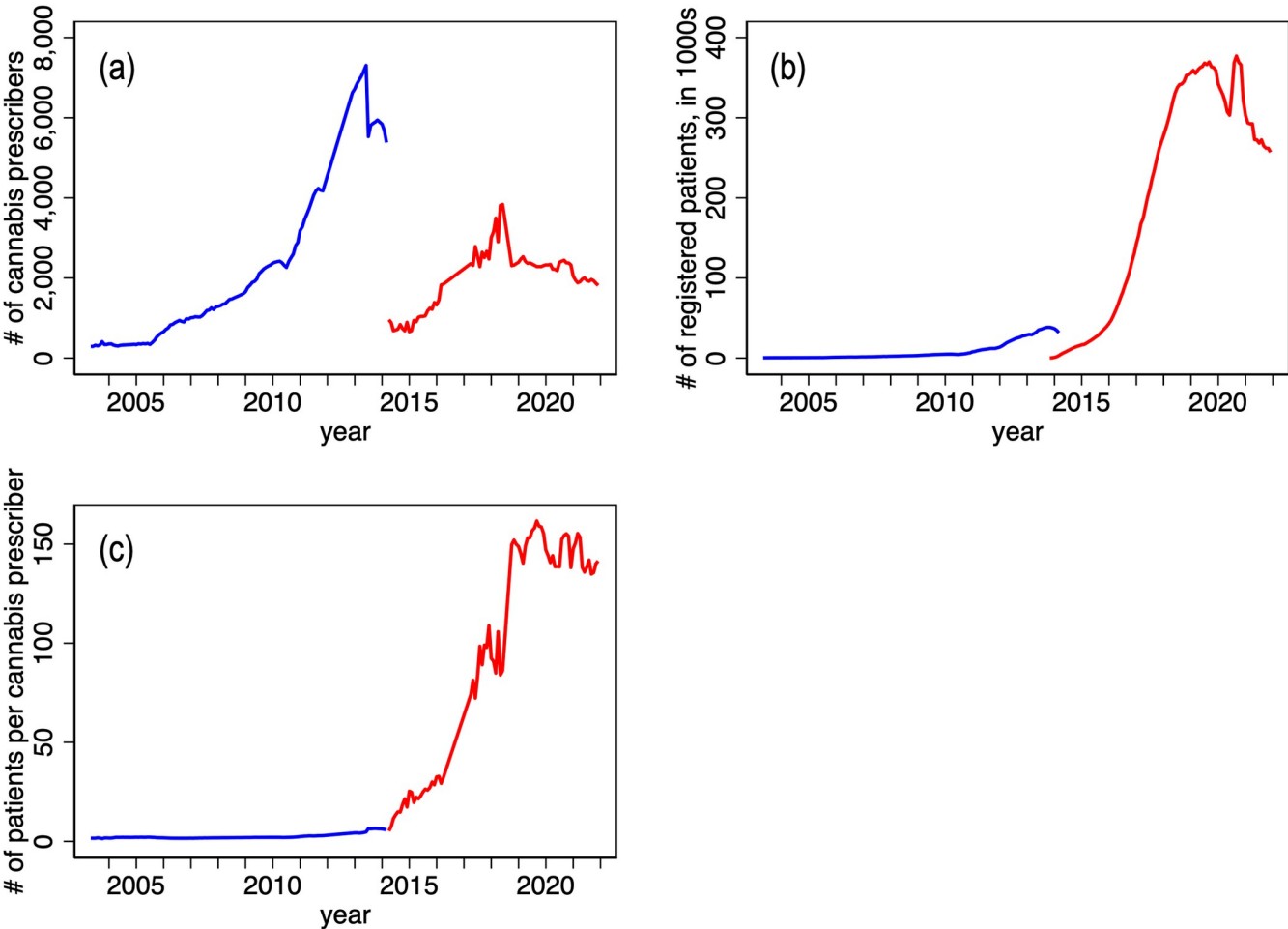

**Fig 2.** Number of physicians prescribing patient medical cannabis use (panel a), number of authorized patients, in thousands (panel b) and number of patients per physician (panel c), by month, 2003–5 to 2014–3 (MMAR) and 2014–4 to 2021–12 (MMPR/ACMPR). Note: Monthly data from the Marihuana Medical Access Regulations (MMAR) regime in blue. Data from the Marihuana for Medical Purposes Regulations (MMPR) and Access to Cannabis for Medical Purposes Regulations (ACMPR) regimes in red. The vertical line indicates the date of legalization of use of non-medical cannabis (October 2018). Data source: Health Canada administrative data provided to the authors.

Panel (a) of Fig 3 reports the sales volumes, in kilograms, of both dry and oil cannabis, by month over the period April 2014 to September 2019. The data indicate that medical cannabis users were increasingly using cannabis oil instead of dry over this period. Cannabis oil sales grew steadily from their market introduction in early 2016, overtaking dry cannabis sales in early 2017, and grew until mid-2019. Unit sales declined in the final two months. Panel (b) of Fig 3 measures sales to consumers in "packaged units" of dry cannabis, cannabis extracts (previously labeled as oils) and edibles over the period October 2019 to March 2022. Medical cannabis users are increasingly using edibles (cannabis-infused drinks and foods). Measured in packaged units, unit sales of extracts, dry cannabis and edibles had roughly equal market shares by March 2022.

There are no statistics on the potency of the cannabis authorized by physicians under the MMPR/ACMPR, but this can be inferred from our survey of Licensed Producer product catalogs. We collected 18,703 observations on dry cannabis products and 6,939 observations on cannabis oil products collected monthly from June 2016 to December 2022. Fig 4 graphs the share of products on offer by Licensed Producers by concentration of THC and CBD and by

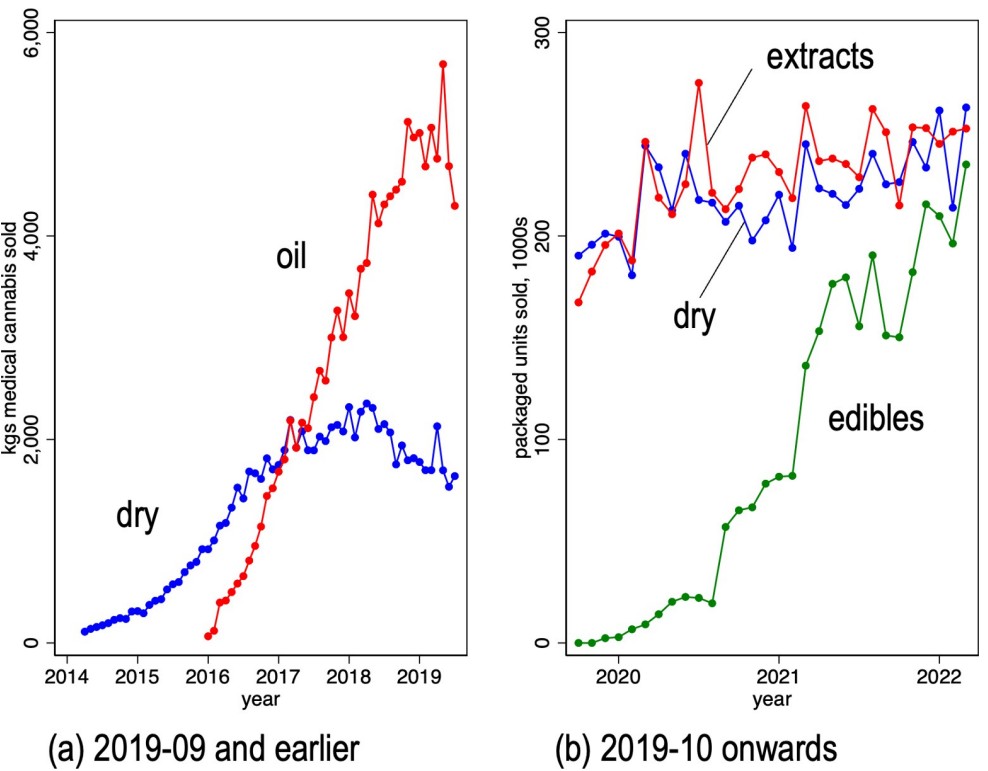

**Fig 3. Volume of cannabis sales from Licensed Producers to registered patients, by cannabis presentation and month.** Panel (a) Volume in kilograms; dry and oil-based presentations; 2014–04 to 2019–09. Panel (b) Volume in packaged units; dry, extracts, and edible presentations; 2019–10 to 2022–03. Panel (a) data source: Health Canada administrative data provided to the authors. Note: after September 2018, cannabis oil sales volumes were measured in litres, not kilograms. Panel (b) data source: Health Canada administrative data obtained from Government of Canada Open Government website [20].

presentation (cannabis oil, dried cannabis) and by month. Among the varieties of cannabis oil on offer, the product share of low THC—high CBD varieties is increasing and now account for the largest product share, slightly higher than the low CBD—high THC products. The latter group of products have been steadily declining over time. Among dry cannabis varieties, over 80% of products are high THC—low CBD and this share is increasing. The median THC content of the dry cannabis varieties available for sale in December 2022 is 20.5% and increasing over time. The median CBD content of the cannabis oil available for sale is about 13% and is also increasing.

## Cannabis prices

Fig 5 reports the inflation-adjusted (i.e., "real") pre-tax price trends of medical cannabis on offer from Licensed Producers by THC and CBD strength and by presentation. Panel (a) displays graphs for mean prices for dry cannabis; panel (b) displays graphs for cannabis oil. Real prices of dried cannabis have tended downwards during the last 3 years. The high THC, high CBD varieties are the exception; these prices have recently increased. The prices of most cannabis oil varieties have also declined in recent years. The exception are the prices for low THC, low CBD varieties.

We next compare changes in Statistics Canada's national medicinal cannabis real price index with changes in the prices of cannabis displayed above. The presentation and dosage

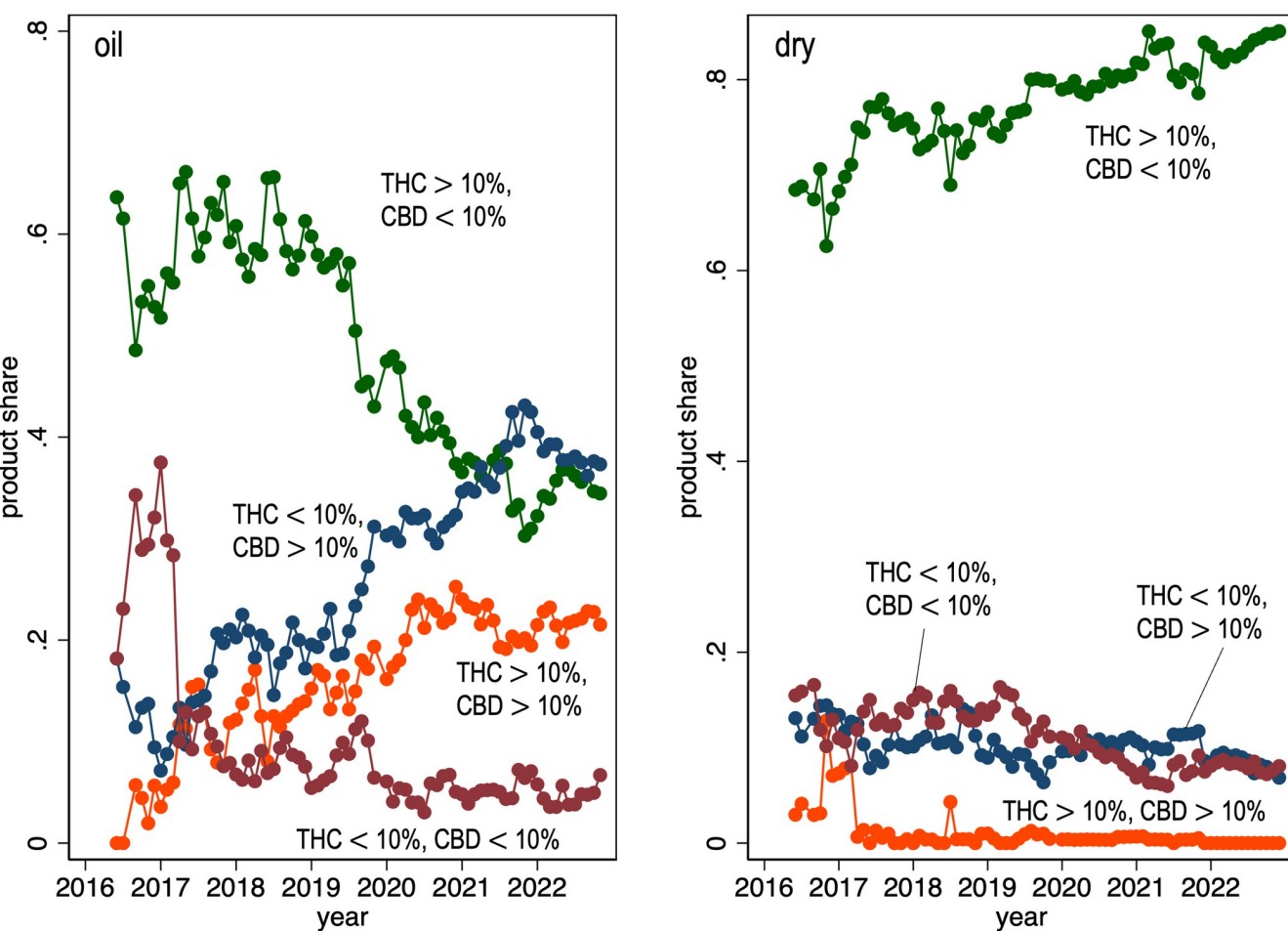

**Fig 4. Fraction of cannabis products sold by Licensed Producers, by THC and CBD concentrations, month (2016–06 to 2022–12) and presentation: Oil vs dry cannabis.** Data Source: monthly review of on-line Licensed Producers product catalogs.

strength specific real prices we collected dropped to varying degrees between January 2019 and October 2021: prices of high THC, low CBD dry varieties dropped by 21%. The low THC, high CBD oil prices dropped by 17% while the high THC, low CBD oil prices dropped by 42%. Statistic Canada's price index declined by 32.4% over the same period, which is within the range of the price declines we found.

## Province-level analysis of patient participation in the medical access program

We next focus on province-specific trends in patient participation in the medical access program. Table 1 presents data on the number of enrollees, number of enrollees with cannabis production licenses, and number of cannabis plants authorized under the MMAR, in December 2013, by province. Table 1 also displays the number of patients, production licenses and plants per 100,000 population aged 20 and older.

Rates of cannabis use authorized under the MMAR program were particularly high in British Columbia. The number of enrollees per capita in that province were about double the rates in Nova Scotia, the next highest province. The number of authorized cannabis plants per capita in British Columbia was about 6 times the rate in Manitoba, the next highest province.

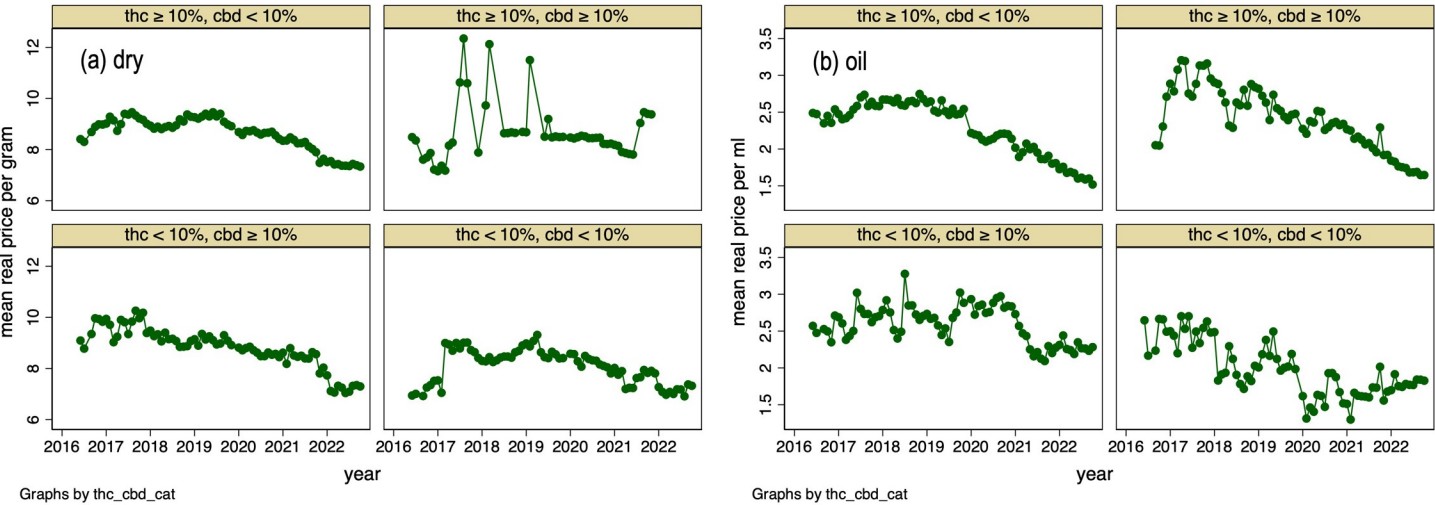

**Fig 5.** Mean real price of cannabis sold by Licensed Producers, by THC and CBD concentration, month (2016–06 to 2022–12) and presentation: dry (a) and oil (b). Data Source: monthly review of on-line product catalogs of Licensed Producers. Note: nominal prices deflated using all-item CPI with 2016–07 = 100. Prices do not include sales and excise taxes.

The MMPR regime, in effect from August 2013 to August 2016, banned personal and delegated cannabis production. Patients were required to obtain cannabis via mail order from Licensed Producers, until the ACMPR replaced the MMPR in 2016. Fig 6 displays the number of cannabis shipments from Licensed Producers per 1,000 population, by month, and patient province of residence. This graph covers the start of the MMPR regime in 2013 to March 2022.

The data indicate that medical cannabis consumption varied markedly across the provinces. Licensed Producer shipments per capita were particularly high in Alberta and Nova Scotia and particularly low in Quebec and British Columbia. The 2016 policy change that allowed for personal or delegated cultivation of medical cannabis did not appear to have an appreciable effect on rates of Licensed Producer shipments. Shipments dropped off in all provinces, to varying degrees, after the legalization of non-medical cannabis in late 2018. This suggests that some medical cannabis program enrollees elected to acquire cannabis

**Table 1. Number of patients, number of patients with cannabis production licenses, and number of cannabis plants authorized under the MMAR, December 2013, by province.**

| Province | # authorized patients | # patients authorized to grow | # plants allowed | # auth patients per 100,000 pop 20+ | # auth to grow per 100,000 pop 20+ | # auth plants per 100,000 pop 20+ | # plants per authorized grower |
|---|---|---|---|---|---|---|---|
| British Columbia | 18,383 | 16,010 | 2,090,743 | 501 | 436 | 56,955 | 131 |
| Alberta | 2,364 | 1,328 | 151,446 | 79 | 44 | 5,048 | 114 |
| Saskatchewan | 952 | 423 | 20,249 | 116 | 52 | 2,469 | 48 |
| Manitoba | 855 | 735 | 82,059 | 91 | 78 | 8,735 | 112 |
| Ontario | 11,071 | 7,322 | 526,242 | 106 | 70 | 5,044 | 72 |
| Quebec | 1,120 | 891 | 78,826 | 18 | 14 | 1,233 | 88 |
| New Brunswick | 805 | 609 | 17,781 | 133 | 101 | 2,945 | 29 |
| Nova Scotia | 1,903 | 1,443 | 40,790 | 254 | 192 | 5,434 | 28 |
| Prince Edward Island | 78 | - | 741 | 70 | - | 661 | - |
| Newfoundland | 192 | 68 | 2,240 | 45 | 16 | 531 | 33 |

Data Source: # cannabis plants authorized: [1]; # patients authorized to grow: [14]; population estimates: [16].

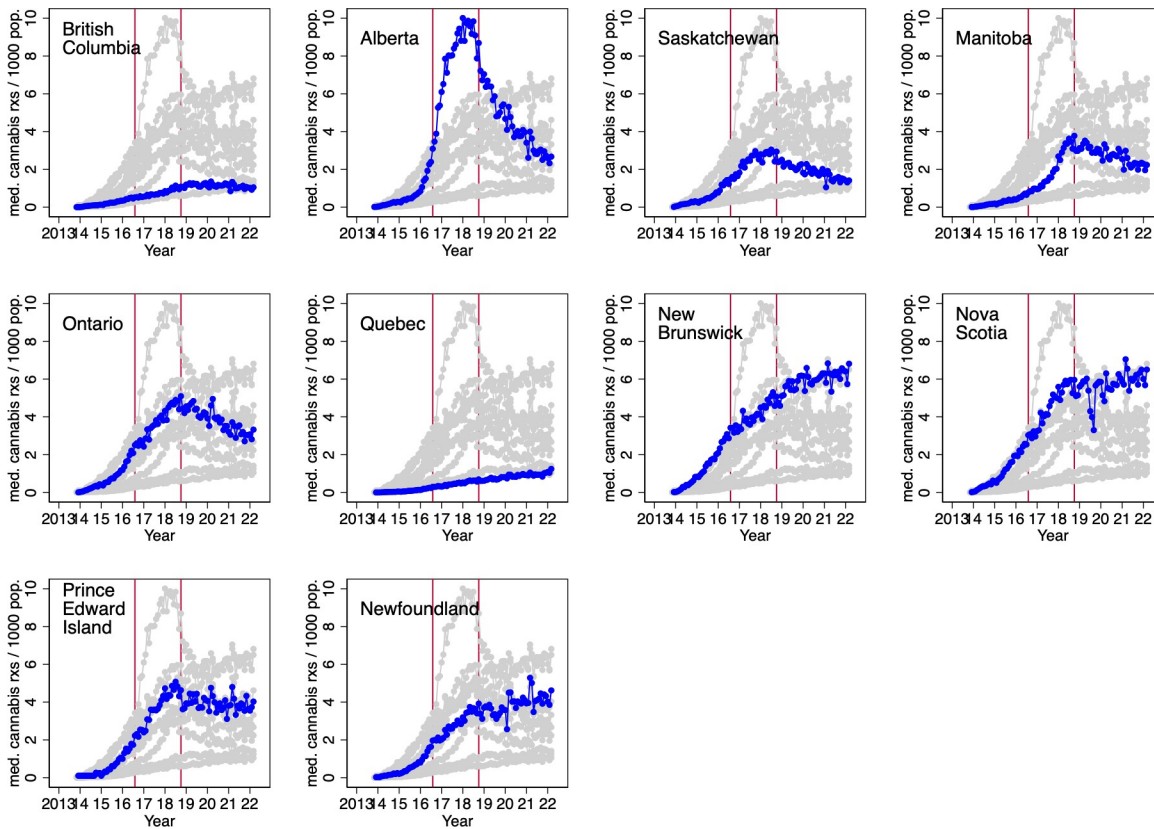

**Fig 6. Number of cannabis shipments from Licensed Producers per 1,000 population, by month, 2013–08 to 2022–03, and patient province of residence.** Note: The first red vertical line indicates the date of introduction of the Access to Cannabis for Medical Purposes Regulations (August 2016), which allowed those authorized to use medical cannabis to cultivate cannabis for personal use; the second red vertical line indicates the date of legalization of use of non-medical cannabis (October 2018). Data source: Health Canada administrative data provided to the authors.

outside of the medical access program, likely from the new retail channels allowed under the recreational cannabis program.

## Impact of province-level factors on medical cannabis use under the MMPR and ACMPR

As was noted, the use of medical cannabis (as measured by shipments from Licensed Producers) varied considerably across provinces under the MMPR and ACMPR regimes. For instance, in January 2018, the per capita number of shipments in Alberta was over ten times the number in the neighbouring province of British Columbia. One potential reason for this variation was the willingness of physicians in the different provinces to authorize cannabis use. This willingness, in turn, could have been affected by the restrictions that the provincial Colleges imposed on cannabis prescribers. To investigate, we graphed province-specific medical cannabis use and the composite cannabis prescribing restriction score as of April 2016 (described in Table 2, below). Medical cannabis use was measured using the average monthly shipments from Licensed Producers per 100,000 population, March to May 2016.

The scatterplot (Fig 7) reveals that per capita medical cannabis shipment volume tended to be lower in provinces with stricter College restrictions on cannabis prescribing.

**Table 2. Provincial College of Physicians restrictions on prescribing of medical cannabis, by province, April 2016.**

| prescribing physician must . . . | province | | | | | | | | | |
|---|---|---|---|---|---|---|---|---|---|---|
| | bc | ab | sk | mb | on | pq | nb | ns | pe | nf |
| meet with patient every 3 months | | 3 | | | | 3 | | | | |
| register with regulator as cannabis prescriber | | 2 | | | | | | | | |
| review patient's medicines use | 2 | 2 | | | | | | | | |
| send in Medical Document to LP | | | | | | 1 | | | | |
| be patient's primary physician | | | 3 | 3 | | | | | | |
| meet patient in person to prescribe | 1 | | | | | | | 1 | 1 | |
| have patient sign written treatment agreement | | | 1 | | 1 | 1 | | | | |
| have patient enroll in research study | | | | | | 2 | | | | |
| create and maintain registry of patients using cannabis | | | | | | 4 | | | | |
| have patient sign consent form | 1 | | | | | 1 | | | 1 | |
| assess patient risk of addiction using standardized tool | 1.5 | 1.5 | | | 1.5 | | | | | 1.5 |
| implement process to identify patient misuse | 2 | 2 | | 2 | 2 | | | | 2 | 2 |
| keep medical documents separate for inspection by regulator | | | 1 | 1 | | 1 | | | | 1 |
| specify THC percentage on Medical Document | | | | | 0.5 | | | | | |
| specify medical condition on Medical Document | | 0.5 | 0.5 | 0.5 | | | | | | |
| **composite cannabis prescribing restriction score** | **7.5** | **11** | **5.5** | **6.5** | **5** | **13** | **0** | **1** | **4** | **4.5** |
| *Higher scores indicate more restrictions* | | | | | | | | | | |

Data Source:

British Columbia: College of Physicians and Surgeons of British Columbia–Practice Standard: Cannabis for Medical Purposes: https://www.cpsbc.ca/files/pdf/PSG-Cannabis-for-Medical-Purposes.pdf

Alberta: College of Physicians and Surgeons of Alberta (BPSA)–Cannabis for Medical Purposes Standard of Practice: https://cpsa.ca/physicians/standards-of-practice/cannabis-for-medical-purposes/

Saskatchewan: College of Physicians and Surgeons of Saskatchewan–Medical Cannabis: https://www.cps.sk.ca/imis/CPSS/CPSS/Programs_and_Services/Medical_Marijuana/Medical_Cannabis.aspx

Manitoba: College of Physicians and Surgeons of Manitoba–Standard of Practice Authorizing Cannabis for Medical Purposes: http://www.cpsm.mb.ca/assets/Standards%20of%20Practice/Standard%20of%20Practice%20 Authorizing%20Cannabis%20for%20Medical%20Purposes.pdf

Ontario: College of Physicians and Surgeons of Ontario–Cannabis for Medical Purposes: https://www.cpso.on.ca/Physicians/Policies-Guidance/Policies/Cannabis-for-Medical-Purposes

Quebec: College des Medecins du Quebec–Ordonnance de Cannabis a des Fins Medicales: http://www.cmq.org/publications-pdf/p-1-2018-09-20-fr-ordonnance-cannabis-fins-medicales.pdf

New Brunswick: College of Physicians and Surgeons of New Brunswick–Guidelines: Medical Marijuana: https://cpsnb.org/en/medical-act-regulations-and-guidelines/guidelines/444-medical-marijuana

Nova Scotia: College of Physicians and Surgeons of Nova Scotia–Professional Standards Regarding the Authorization of Marijuana for Medical Purposes: https://cpsns.ns.ca/wp-content/uploads/2017/12/Authorization-of-Marijuana-Medical-Purposes.pdf

Prince Edward Island: The College of Physicians and Surgeons of Prince Edward Island–Prescribing of Medical Marijuana Policy: https://cpspei.ca/wp-content/uploads/2017/03/Marijuana-Prescribing-Nov-3016.pdf

Newfoundland and Labrador: College of Physicians and Surgeons of Newfoundland and Labrador–Medical Marihuana: Advisory and Interim Guideline: https://imis.cpsnl.ca/WEB/CPSNL/Policies/Advisory_and_Interim_Guideline_-_Medical_Marihuana.aspx

One possible reason for this association is that rates of regular cannabis use–both medical and non-medical–are lower in provinces with stricter College regulations. However, Fig 8 reveals that the estimated fraction of regular users who are enrolled in the medical access program also tended to be higher in provinces with less stringent prescribing controls. Thus, it appears that the cannabis prescribing restrictions affected rates of participation in the medical cannabis program.

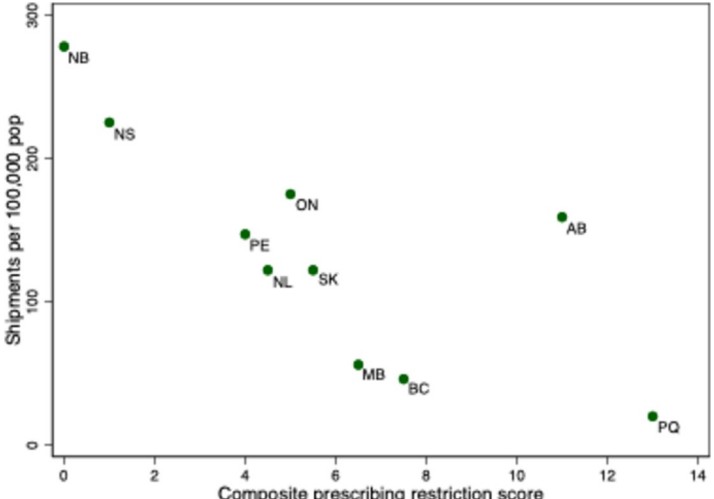

**Fig 7. Average monthly cannabis shipments from Licensed Producers per 100,000 population, March to May 2016, and composite cannabis prescribing restriction score, by province.** Note: BC = British Columbia, AB = Alberta, SK = Saskatchewan, MB = Manitoba, ON = Ontario, QC = Quebec, NB = New Brunswick, NS = Nova Scotia, PE = Prince Edward Island, NF = Newfoundland and Labrador. Shipments Data Source: Health Canada administrative data provided to the authors.

Fig 7 also suggests that some variation in the volume of Licensed Producer shipments was due to factors other than prescribing restrictions. In particular, British Columbia and Manitoba had particularly low shipment volumes, and Alberta, high volumes, given their prescribing restriction score. The low volume of Licensed Producer shipments to British Columbia and Manitoba could have been due to medical cannabis users in these provinces

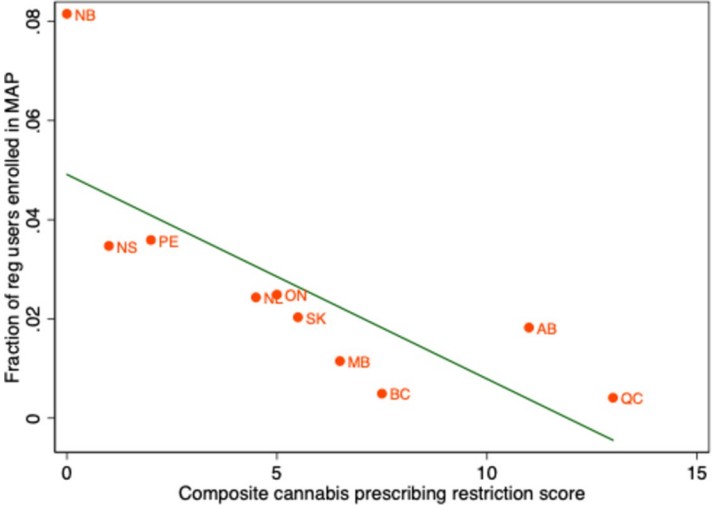

**Fig 8. Ratio of patients enrolled in medical cannabis access program to the estimated population aged 20 years and older that uses cannabis daily or weekly, by province, July 2015.** Data Source: Enrollment in medical access program (MAP): Health Canada administrative data provided to the authors. Estimated population aged 20 years and older that uses cannabis daily or weekly: 2015 Statistics Canada Canadian Tobacco, Alcohol and Drugs Survey. BC = British Columbia, AB = Alberta, SK = Saskatchewan, MB = Manitoba, ON = Ontario, QC = Quebec, NB = New Brunswick, NS = Nova Scotia, PE = Prince Edward Island, NL = Newfoundland and Labrador.

accessing cannabis from plants that were established under production licenses in the previous MMAR regime.

### Rates of regular cannabis use in the community over time

As Table 1 illustrates, during the MMAR regime, almost 3 million cannabis plants were licensed for cultivation in residences. Fig 2 illustrates a marked increase in the number of enrollees in the medical cannabis program across most provinces after eligibility requirements were relaxed in 2013. There was a concurrent increase in medical cannabis sales volumes, from about 300 kg/month in early 2015 to over 6000 kg/month in mid 2018 (Fig 3). This liberalization of the medical cannabis access program may have increased awareness of cannabis' therapeutic uses and reduced the social stigma from its use. Home-based cultivation, followed by the creation of the Licensed Producer industry may have also increased the availability of cannabis through unauthorized channels, such as through dispensaries and online dealers.

To investigate, we examined rates of regular (daily or almost daily) cannabis use among each provincial adult population from 1989 to 2021; this period spans the different medical cannabis access program regimes and the 2018 legalization of recreational cannabis. These estimates were obtained from the various national health and drug use surveys conducted over this period; we used the sampling weights provided in each survey to render the estimates representative of the residential adult population.

The following features are evident from the province-specific graphs displayed in Fig 9. First, rates of regular cannabis use increased in all provinces over the three decades. Second, the introduction of the 2003 MMAR did not appear to result in an increase in regular cannabis use. However, the 2014 MMPR regime appeared to increase rates of regular use in all provinces, except for Quebec. Finally, the legalization of recreational cannabis in 2018 appeared to markedly increase daily cannabis use in some provinces.

### Rates of regular cannabis use in the community and participation in the medical access program over time

Finally, we compared the national participation rates in the medical access program from 2005 to 2021 to the estimated rates of daily cannabis use over the same period (Fig 10). The difference between these reflects the estimated fraction of the adult population that uses cannabis daily outside of the medical cannabis access program and thus likely without physician supervision. We find that only a small fraction of regular users enrolled in the medical access program during the MMAR regime. Rates of enrollment in the medical access program increased under the MMPR, but rates of daily cannabis use increased even faster. Thus, the fraction of daily users who did so outside of the medical access program increased under the MMPR regime. This fraction increased again after the legalization of recreational cannabis use in 2018.

### Discussion

This study is the first to examine long-term trends in Canada's medical cannabis sector and to relate these trends to the medical and recreational cannabis regulatory policy changes. Health Canada initially restricted access to those who had the personal approval of the federal Minister of Health. The courts ruled that the approvals process infringed patients' constitutional rights [2]. Health Canada, in turn, launched a medical cannabis access program in 2001 that granted access to those with physician-documented evidence of a severe health problem that could not be managed using conventional therapies. Most patients accessed cannabis grown under a personal or delegated production license. By 2013, almost 3 million plants were

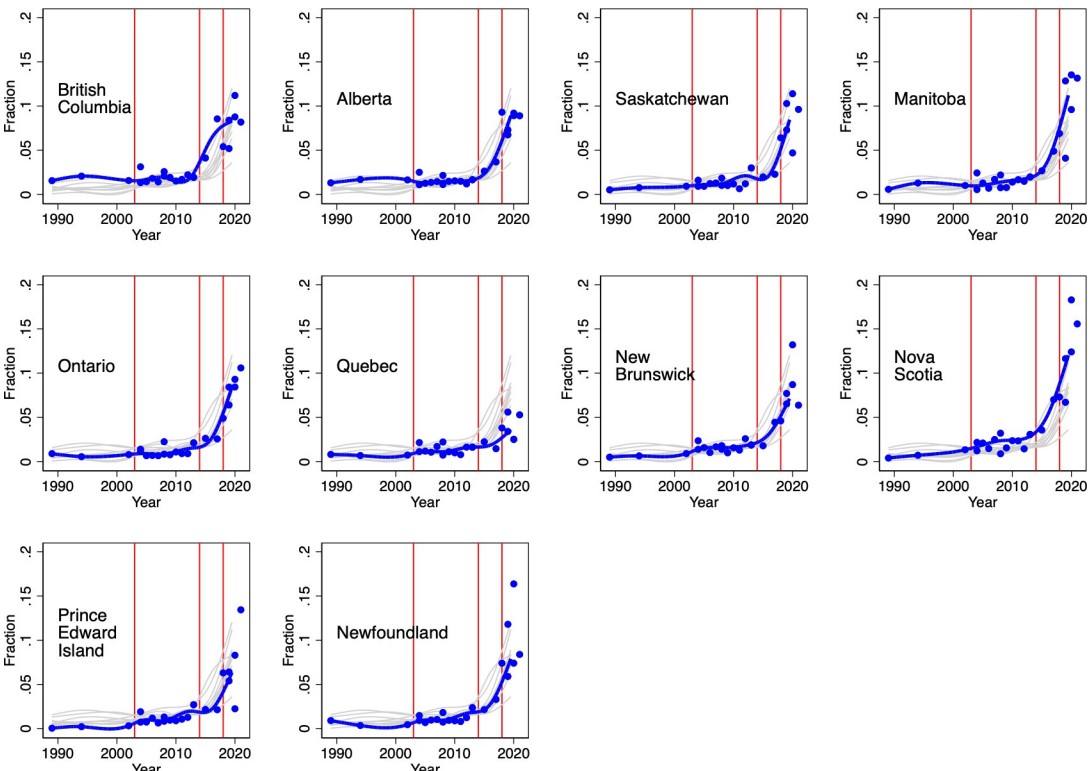

**Fig 9. Estimated fraction of community dwelling population aged 20 years and older that consumes cannabis daily or almost daily, by province, survey, and year, 1989–2021.** Note: the blue line is a median spline curve fitted to the provincial estimates (the blue dots). The red vertical lines indicate the introduction of the following policies: 1) the 2001 Marihuana Medical Access Regulations (MMAR) regime which provided medical cannabis access to severely ill patients; 2) the 2014 Marihuana for Medical Purposes Regulations (MMPR) regime which delegated medical cannabis authorization to physicians and expanded the indications for cannabis use; and 3) the 2018 legalization of recreational cannabis. Data source: 1989 National Alcohol and Drugs Survey; 1994 Canada's Alcohol and Other Drugs Survey; 2002 Canadian Community Health Survey; 2004 Canada Addiction Survey (CAS); 2005–2013 Canadian Tobacco Use Monitoring Survey; 2008–2012 Canadian Alcohol and Drug Use Monitoring Surveys (CADUMS); 2013, 2015, 2017 Canadian Tobacco, Alcohol and Drugs Survey; 2019–2021 Canadian Tobacco and Nicotine Survey; and the 2018, 2019 and 2020 National Cannabis Survey (NCS). All surveys were conducted by Statistics Canada except for the 2004 CAS, the 2008–2012 CADUMS, and the 2018–2020 NCS which were commissioned by Health Canada and conducted by private sector survey firms.

authorized and the average authorized daily consumption was about 18 grams. Concerned over production safety and illegal diversion, Health Canada in 2014 required that cannabis be purchased from a Licensed Producer, a regulated commercial grower; personal production was banned. Health Canada also ceded to physicians responsibility for authorizing patient access to cannabis.

To fill the regulatory void, each of the physician Colleges imposed their own prescribing restrictions. The number of physicians who were willing to support patient cannabis use subsequently fell; the physicians that were willing to do so, however, authorized cannabis use for large numbers of patients. The result was a marked increase in the number of patients participating in the program.

The post-2014 increase in medical cannabis use varied by province–rates of medical cannabis use and the fraction of regular cannabis users who participated in the medical cannabis program were generally lower in provinces with stricter regulations on physician cannabis prescribing. This finding is consistent with survey results reported by Belle-Isle and colleagues; patients in some provinces reported difficulty in finding physicians willing to support medical

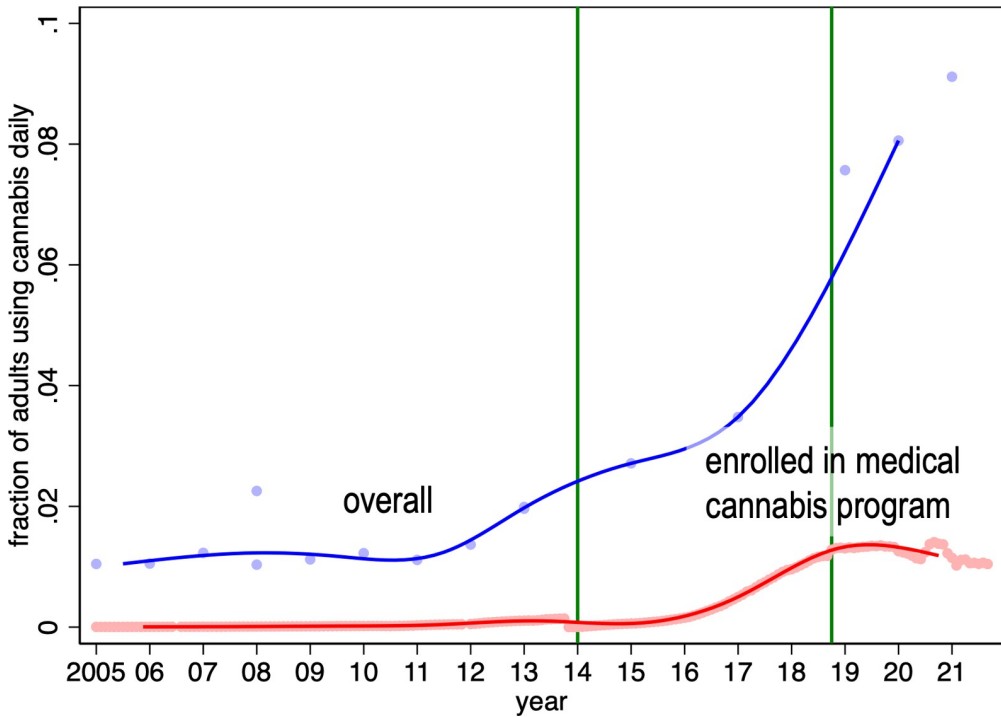

**Fig 10. Estimated fraction of national community dwelling population aged 20 years and older that consumes cannabis daily or almost daily, by month, and estimated fraction of population age 20 years and older enrolled in medical cannabis access program, by month 2005m1 – 2021m12.** Note: the blue line is a median spline curve fitted to the national estimates of fraction of population 20+ that uses cannabis daily (the blue dots). The red line is a median spline curve fitted to the national participation rates in the medical access program (the red dots). The green vertical lines indicate the introduction of the following policies: 1) the 2014 Marihuana for Medical Purposes Regulations (MMPR) regime which delegated medical cannabis authorization to physicians and expanded the indications for cannabis use; and 2) the October 2018 legalization of recreational cannabis.

cannabis use and thus turned to dispensaries and other unsanctioned outlets [27]. It also seems plausible that the low rate of program participation in British Columbia was due to the availability of cannabis from dispensaries. In less populous regions of the province, in which the federal police force, the Royal Canadian Mounted Police (RCMP), provided local law enforcement, illicit cannabis distribution was forced to go underground. However, law enforcement agencies in some large cities in the province allowed cannabis dispensaries to operate [28]. The City of Vancouver even regulated cannabis dispensaries, despite their illegality at the time [29]. Province-wide, the average penalty issued for cannabis possession in BC was lower than in other provinces [30]. This suggests that some of the interprovincial variation in the participation in the MMPR regime was due to variation in the propensity of authorities to enforce the CDSA.

It appears that patients who participate in the medical access program increasingly are using cannabis oils instead of dried cannabis. This accords with the results of surveys of individuals who report using cannabis for medical reasons [31–33]. Most varieties of cannabis oil sold by Licensed Producers are now high in CBD and low in THC. Dry cannabis varieties, conversely, tend to be high in THC and low in CBD.

A key policy question is the price spread between medical cannabis grown by Licensed Producers and cannabis available from unauthorized sources. Medical cannabis is typically more expensive because it is subject to excise and sales taxes and must meet stringent production safety requirements. Ideally, the price difference between authorized and unauthorized

cannabis is sufficiently small such that consumers would select the former source. We find that real prices of most varieties of medical cannabis are declining over time. However, there is little information on changes in the illicit market prices for the same varieties favored by medical cannabis users; the available evidence focuses on prices of dry cannabis [34, 35].

We estimated rates of daily cannabis use (medical use or otherwise) by province using national surveys conducted over the last three decades and find that rates increased markedly after the introduction of the 2014 MMPR policy regime. Only a small fraction of daily cannabis users (<10%) were enrolled in the medical cannabis access program. The fraction of Canadians using cannabis daily increased markedly after the 2018 legalization of recreational cannabis; at the same time, participation in the medical access program declined. This suggests that increasing numbers of Canadians are using cannabis daily without medical supervision. The literature indicates that individuals who use cannabis therapeutically use it primarily to manage pain and insomnia, and to alleviate the symptoms of psychiatric disorders (such as anxiety and depression) [33, 36, 37]. The health outcomes of the growing numbers of patients using cannabis regularly–and evidently without physician supervision–remains an important area for future research.

## Acknowledgments

We thank two anonymous referees for helpful comments and thank Jonathan Fortin at Health Canada for supplying administrative data. We acknowledge the research assistance of Igor Korolija.

## Author Contributions

**Conceptualization:** Paul Grootendorst.

**Data curation:** Minsup Shim.

**Investigation:** Hai Nguyen, Paul Grootendorst.

**Visualization:** Minsup Shim, Paul Grootendorst.

**Writing – original draft:** Paul Grootendorst.

**Writing – review & editing:** Minsup Shim, Hai Nguyen, Paul Grootendorst.

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
