## [Decision Letter · Decision Letter 0]

10 Oct 2022

PONE-D-22-17767Lessons from 20 years of medical cannabis use in CanadaPLOS ONE

Dear Dr. Grootendorst,

Thank you for submitting your manuscript to PLOS ONE. After careful consideration, we feel that it has merit but does not fully meet PLOS ONE’s publication criteria as it currently stands. Therefore, we invite you to submit a revised version of the manuscript that addresses the points raised during the review process.

We urge you to pay particular attention to the concerns of reviewer 2 regarding the data handling, transparency, and accessibility. Further, reviewer 1 notes that the manuscript needs to be revised to be more accessible to a non-Canadian (international) audience. Overall, both reviewers raise a number of substantive concerns.

We look forward to receiving your revised manuscript.

Kind regards,

Julian Aherne

Academic Editor

PLOS ONE

Journal Requirements:

Additional Editor Comments (if provided):

I have received comments from two reviewers, they waiver between major revisions and reject but note that it is a “potentially interesting and useful paper”. I have recommended major revisions under the assumption that a revised manuscript can adequately address all reviewers’ concerns. I urge the authors to pay particular attention to the concerns of reviewer 2 regarding the data handling, transparency, and accessibility. Further, reviewer 1 notes that the manuscript needs to be revised to be more accessible to a non-Canadian (international) audience. Please provide a point-by-point response to each reviewers comment and indicate how the manuscript has been revised.

Reviewers' comments:

Reviewer's Responses to Questions

**Comments to the Author**

1. Is the manuscript technically sound, and do the data support the conclusions?

Reviewer #1: Partly

Reviewer #2: Yes

2. Has the statistical analysis been performed appropriately and rigorously? 

Reviewer #1: Yes

Reviewer #2: N/A

3. Have the authors made all data underlying the findings in their manuscript fully available?

Reviewer #1: Yes

Reviewer #2: No

4. Is the manuscript presented in an intelligible fashion and written in standard English?

Reviewer #1: Yes

Reviewer #2: Yes

5. Review Comments to the Author

Reviewer #1: This is a potentially interesting and useful paper on the evolution of Canada’s medical cannabis system during the past two decades. The paper points to an interesting story about the interplay of changes in policy at the federal and provincial levels and how those change were reflected in amount of cannabis produced and consumed, the nature of cannabis products, and price of cannabis. The paper would be improved by some edits to help make the paper more accessible to a non-Canadian audience and more clearly describing variation and changes provincial policies. Below are specific suggestions.

1. Not being Canadian, I was unsure what Health Canada and physician Colleges are. The authors should make edits in the Abstract and throughout the paper to make the paper more accessible to a non-Canadian audience. I believe that physician Colleges set province-specific guidelines for medical cannabis, but that is not clearly stated anywhere in the paper.

2. The interplay between provincial and federal regulations could perhaps be discussed more clearly. Pre-2014, it seems there was a huge difference across provinces in authorizations for medical cannabis (lots more medical cannabis authorized in BC), but this changed in 2014 when a system of commercial growers was created. After 2014, it looks like the size of the medical cannabis market was relatively small in BC. I don’t think this dramatic shift is clearly explained in the paper. Maybe there is no easy explanation. There is a story here about variation in policies, regulations, and maybe enforcement of regulations across provinces and how those things evolved across years.

3. The section spanning pages 4 through 6 could be more succinct and would perhaps be helped by a figure showing a policy-change timeline. Ideally, you might have separate timelines for each province as well, although I am not sure that information on changes in province-level policies across years is available to the authors. (They do have information on provincial variation in restrictions in 2016.)

4. The issue of variation in enforcement of cannabis laws might be worth addressing more explicitly. In the US, one aspect of liberalization of cannabis policy is decriminalization, involving reduced priority given to enforcement of cannabis laws as well as reduction in penalties for violation of cannabis laws. I was unable to figure out how much of this is an issue in Canada’s cannabis laws. One guess I have about what happened in BC is that after cannabis growing by those authorized for medical cannabis was banned, very little effort was put into enforcing that ban and people continued to supply themselves by growing their own cannabis (or having a friend who grew their own) rather than switching to authorized commercial growers. There was probably a lot of diversion occurring in BC, although from Figure 8 it does not look like population-wide prevalence of daily or near daily use was much higher in BC than most other provinces.

5. There were some numbers that were surprising to me about the average number of grams per day per medical cannabis user. In dry form, a gram of cannabis is about 3 joints. Someone who uses daily and heavily might use 1 to 2 grams per day. I was surprised to see 18 grams per day allowed under MNAR in 2013 (page 5) or the average daily dose of cannabis authorized by Health Canada of 5.6 grams (page 10). Even 2 grams per day prescribed by physicians under MMPR (page 11) seems awfully high. Am I misunderstanding something?

6. The authors put information on variation in medical cannabis restrictions by province in 2016 in an online supplement. I would like to see this table included in the body of the paper.

7. In examining national survey data on cannabis use, the authors focus on daily or near daily use. What was the cutoff for daily use? A common cutoff is 20+ days of use in the past 30. Is that what was used here?

8. The authors provide information on price and potency separately for dry cannabis and oils. The authors should briefly clarify how these are linked to different routes of administration (e.g., smoking, vaping, dabbing, and edibles).

9. Does Figure 4 correct for inflation?

Reviewer #2: Uploaded letter, attached.

The authors provide an extensive descriptive history of the Canadian medical cannabis regime by collating data from a wide variety of sources to provide an integrated overview of how the sector has evolved since inception. An extensive amount of data is processed to provide statistical snapshots of several dimensions of the Canadian medical cannabis system over time, including after legalization. The analysis I believe is mostly novel, the manuscript is well-written if not always technically clear, and there are no statistical errors that I can identify.

Unfortunately I can only recommend that PLOS reject the manuscript. The data handling, transparency, and accessibility clearly do not meet the standards of PLOS. This could be potentially fixed, which makes a case for major revisions, but my reading of the manuscript and the authors’ declarations suggests they will be unable to meet the PLOS data standards and hence I recommend rejection. I lay out a number of crucial issues leading to this recommendation below.

6. PLOS authors have the option to publish the peer review history of their article (what does this mean?). If published, this will include your full peer review and any attached files.

Reviewer #1: No

Reviewer #2: No

---

## [Author Response · Author response to Decision Letter 0]

8 Dec 2022

Responses appear in the file "Response to PLOS Reviews" which I have uploaded.

---

## [Decision Letter · Decision Letter 1]

7 Feb 2023

PONE-D-22-17767R1Lessons from 20 years of medical cannabis use in CanadaPLOS ONE

Dear Dr. Grootendorst,

Thank you for submitting your manuscript to PLOS ONE. After careful consideration, we feel that it has merit but does not fully meet PLOS ONE’s publication criteria as it currently stands. Therefore, we invite you to submit a revised version of the manuscript that addresses the points raised during the review process.

Please ensure that the revised manuscript, including the abstract, is accessible to an international audience; provide clarification regarding the number of grams authorized; and give the manuscript a thorough edit to increase clarity, e.g., ensure manuscript is written entirely in the past tense.

We look forward to receiving your revised manuscript.

Kind regards,

Julian Aherne

Academic Editor

PLOS ONE

Journal Requirements:

Additional Editor Comments:

Based on the comments from one reviewer, the manuscript requires minor revisions: please ensure that the paper, including the abstract, is accessible to an international audience; provide clarification regarding the number of grams authorized; and

lastly the manuscript requires a thorough edit to increase clarity, e.g., ensure manuscript is written entirely in the past tense. I look forward to your revised submission.

Reviewers' comments:

Reviewer's Responses to Questions

**Comments to the Author**

1. If the authors have adequately addressed your comments raised in a previous round of review and you feel that this manuscript is now acceptable for publication, you may indicate that here to bypass the “Comments to the Author” section, enter your conflict of interest statement in the “Confidential to Editor” section, and submit your "Accept" recommendation.

Reviewer #1: (No Response)

2. Is the manuscript technically sound, and do the data support the conclusions?

Reviewer #1: Partly

3. Has the statistical analysis been performed appropriately and rigorously? 

Reviewer #1: Yes

4. Have the authors made all data underlying the findings in their manuscript fully available?

Reviewer #1: Yes

5. Is the manuscript presented in an intelligible fashion and written in standard English?

Reviewer #1: Yes

6. Review Comments to the Author

Reviewer #1: The revised manuscript includes additional information requested by the other reviewer and me. The authors’ responses to our question and suggestions were thoughtful. I continue to think the information on the evolution of medical cannabis is Canada presented in the manuscript will be useful to the field. The authors do not run statistical models assessing statistical significance of associations or estimating magnitude of “effects” of policy changes. I think that is fine, although there are place in the paper where the authors should make edits to be more cautious about causal inferences.

The other reviewer raised concerns about data sharing and transparency. My understanding is that almost all the data used by the authors is already publicly available. Where they have put effort into interpreting or coding that data—e.g., rating restrictiveness of province policies—they provide information on their process for doing so. I am not entirely clear on all the rules for data sharing in PLOS ONE publications, but it seems like the authors are in accord with the spirit of transparency.

I had requested the authors make sure their paper was accessible to an international audience. The authors added some clarification on how the system of physician Colleges works in setting province-specific policies into the main body of the paper. It would help readers if the Abstract was written in a way that made it clear what Health Canada and physician Colleges are.

Both the other reviewer and I reacted to information reported on average daily doses of cannabis. The response letter provided some reasons why the average authorized dose under MMAR would have been so high. My takeaway from the authors response is that the number of grams authorized under MMAR, the average number of grams supplied under MMPR, and the actual number of grams per day consumed by medical cannabis users are very different things. It would help the reader if the authors added some clarifying information to the text of the manuscript. If they don’t, many readers, thinking in terms of actual grams consumed, are going to think that the high numbers are due to typographical or data coding errors.

In general, the writing quality of the paper is fine in terms of clarity. That said, this paper presents a lot of information and, if you are not Canadian, a lot of that information is new and confusing. The content of the paper also does not neatly fit into the conventional format of journal article. For example, should history of medical cannabis policy be in the Results section of the Abstract? The structure of the Abstract and the body of the manuscript do not match. The paper could be published in close to its current form, but I believe the authors would be much happier with their paper, and it would receive a much wider audience, if it received a thorough edit to increase clarity, eliminate some typographical and grammatical errors, increase precision of word choice, and, in general, make the paper more reader friendly. One recommendation I have is that the paper be written almost entirely in the past tense. There are several places where the authors slip into the present tense, which tends to impart stronger inferences than the past tense. The strength of inferences is general issue with this paper. The manuscript is primarily descriptive. I would caution against make statement that imply specific policy changes “caused…”

7. PLOS authors have the option to publish the peer review history of their article (what does this mean?). If published, this will include your full peer review and any attached files.

Reviewer #1: No

---

## [Author Response · Author response to Decision Letter 1]

8 Mar 2023

In this revision, 

• we have revised the abstract as requested. 

• we now clarify that MMAR program enrollees likely did not consume all the authorized cannabis dosage amounts (especially given the concerns that cannabis was being diverted). 

• we removed any statements that implied that a specific policy change exerted a causal impact on the outcome variables studied in this paper.

• we have also thoroughly edited the manuscript to improve clarity.

---

## [Editor Report · Decision Letter 2]

9 Mar 2023

Lessons from 20 years of medical cannabis use in Canada

PONE-D-22-17767R2

Dear Dr. Grootendorst,

We’re pleased to inform you that your manuscript has been judged scientifically suitable for publication and will be formally accepted for publication once it meets all outstanding technical requirements.

Kind regards,

Julian Aherne

Academic Editor

PLOS ONE

Additional Editor Comments (optional):

The revised manuscript addresses the concerns of the reviewers. I recommend that it is accepted for publication.
---

## [Editor Report · Acceptance letter]

15 Mar 2023

PONE-D-22-17767R2 

Lessons from 20 years of medical cannabis use in Canada 

Dear Dr. Grootendorst:

I'm pleased to inform you that your manuscript has been deemed suitable for publication in PLOS ONE. Congratulations! Your manuscript is now with our production department. 

Kind regards, 

on behalf of

Dr. Julian Aherne 

Academic Editor

PLOS ONE